

# North Atlantic Oscillation drives the annual occurrence of an isolated, peripheral population of the brown seaweed *Fucus guiryi* in the Western Mediterranean Sea

Ignacio J. Melero-Jiménez[1], A. Enrique Salvo[1], José C. Báez[2], Elena Bañares-España[1], Andreas Reul[3] and Antonio Flores-Moya[1]

[1] Departamento de Biología Vegetal (Botánica), Facultad de Ciencias, Universidad de Málaga, Málaga, Spain
[2] Centro Oceanográfico de Canarias, Instituto Español de Oceanografía, Santa Cruz de Tenerife, Spain
[3] Departamento de Ecología y Geología, Facultad de Ciencias, Universidad de Málaga, Málaga, Spain

Corresponding author
Antonio Flores-Moya, floresa@uma.es

## ABSTRACT

The canopy-forming, intertidal brown (Phaeophyceae) seaweed *Fucus guiryi* is distributed along the cold-temperate and warm-temperate coasts of Europe and North Africa. Curiously, an isolated population develops at Punta Calaburras (Alboran Sea, Western Mediterranean) but thalli are not present in midsummer every year, unlike the closest (ca. 80 km), perennial populations at the Strait of Gibraltar. The persistence of the alga at Punta Calaburras could be due to the growth of resilient, microscopic stages as well as the arrival of few–celled stages originating from neighbouring localities, and transported by the permanent Atlantic Jet flowing from the Atlantic Ocean into the Mediterranean. A twenty-six year time series (from 1990 to 2015) of midsummer occurrence of *F. guiryi* thalli at Punta Calaburras has been analysed by correlating with oceanographic (sea surface temperature, an estimator of the Atlantic Jet power) and climatic factors (air temperature, rainfall, and North Atlantic Oscillation –NAO-, and Arctic Oscillation –AO- indexes). The midsummer occurrence of thalli clustered from 1990–1994 and 1999–2004, with sporadic occurrences in 2006 and 2011. Binary logistic regression showed that the occurrence of thalli at Punta Calaburras in midsummer is favoured under positive NAO index from April to June. It has been hypothesized that isolated population of *F. guiryi* should show greater stress than their congeners of permanent populations, and to this end, two approaches were used to evaluate stress: one based on the integrated response during ontogeny (developmental instability, based on measurements of the fractal branching pattern of algal thalli) and another based on the photosynthetic response. Although significant differences were detected in photosynthetic quantum yield and water loss under emersion conditions, with thalli from Punta Calaburras being more affected by emersion than those from Tarifa, the developmental instability showed that the population from Tarifa suffers higher stress during ontogeny than that from Punta Calaburras. In conclusion, this study demonstrates the teleconnection between atmospheric oscillations and survival and proliferation of marine macroalgae.

## INTRODUCTION

The canopy-forming, brown (Phaeophyceae) seaweed *Fucus guiryi Zardi et al. (2011)* inhabits the littoral zone of cold-temperate and warm-temperate European and African coasts of the northern Atlantic Ocean. This species is more abundant in upwelling areas, which are considered to be climatic change refugia (*Lourenço et al., 2016*). The known southern limit of distribution occurs in Dakhla, Western Sahara (*Lourenço et al., 2016*), but the range does not extend continuously into the Mediterranean Sea (*Zardi et al., 2011*). However, an isolated population develops at Punta Calaburras (Alboran Sea, Western Mediterranean; Figs. 1A–1B), around 80 km from the nearest populations in the Strait of Gibraltar (*Conde, 1989*). It has been hypothesized that the presence of *F. guiryi* (previously known as *F. spiralis* and *F. spiralis* var. *platycarpus*; see *Zardi et al., 2011*) at Punta Calaburras is favoured by the "Atlantic Jet" current flowing from the Atlantic Ocean into the Mediterranean Sea through the Strait of Gibraltar (*Bellón, 1953*; *Conde & Seoane-Camba, 1982*). Punta Calaburras is located at the edge of the North Western Alboran upwelling (*Reul et al., 2005*; *Muñoz et al., 2015*; *Macías, García-Gorriz & Stips, 2016*), which is in agreement with the hypothesis that upwelling areas provide thermal refugia for *F. guiryi* (*Lourenço et al., 2016*). At Punta Calaburras, the Atlantic Jet (AJ) approaches the Spanish coast before deflecting towards the Moroccan coast (Fig. 1A). This current compensates for the negative water balance in the Mediterranean Sea due to the loss of water by evaporation, which is greater than the inputs by precipitation and rivers (*Rodríguez, 1982*; *Parrilla & Kinder, 1987*). Although *F. guiryi* (as *F. spiralis*; see *Zardi et al., 2011*) was found in 1987 on the Mediterranean coast of France at Gruissan (Aude), it was probably introduced via oyster culture in the lagoons along the Mediterranean French coasts (*Sancholle, 1988*); this introduced population is not comparable to the natural population of Punta Calaburras.

The singularity of the isolated population of *F. guiryi* at Punta Calaburras inspired us to start a survey in 1990 (linked to the field teaching at the university of the corresponding author AFM), which revealed that the population was always detected in winter. However, in some years the thalli did not survive midsummer, in contrast to the nearby perennial populations in the Strait of Gibraltar and throughout the range of this species. The occurrence of this population of *F. guiryi* must be determined by environmental conditions, so a first aim of this study was to analyse the role of the oceanographic and atmospheric factors controlling the presence of thalli at Punta Calaburras in midsummer. For this purpose, the time series (from 1990 to 2015) of presence/absence of *F. guiryi* thalli at Punta Calaburras was analysed by binary logistic regression, using two kinds of independent, explanatory variables. First, a proxy for the powerful incoming current from the Atlantic Ocean into the Alboran Sea, which becomes evident by changes in sea surface temperature (SST; lower SST with higher current flow; *Vargas-Yáñes et al., 2002*; *Renault et al., 2012*).

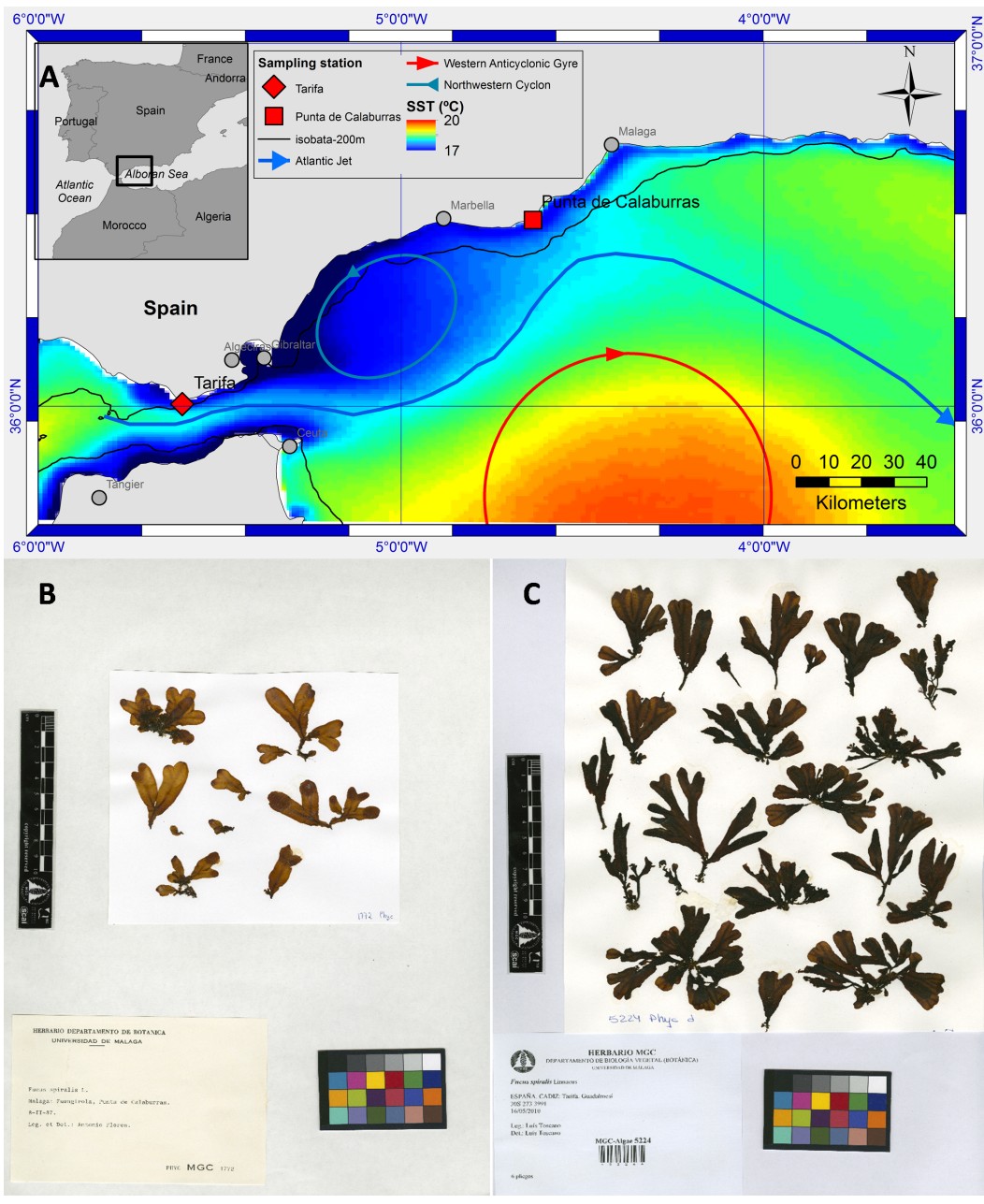

**Figure 1** **Map of study sites, oceanographic conditions and herbarium sheets.** Map of the area near the Strait of Gibraltar (A) showing the two sampling points of *Fucus guiryi* on the SW Iberian Peninsula (Tarifa and Punta Calaburras). Mean Sea Surface Temperature (SST) of weekly SST images 1998–2014 (cold upwelled water in blue (17 °C) and warm water of the Western Alboran Anticyclon in red (20 °C) as well as a schematic circulation pattern. Herbarium sheets of *F. guiryi* collected at Punta Calaburras (B) and Tarifa (C). It must be noted that the samples are identified according to the synonim *Fucus spiralis*.

Second, the North Atlantic Oscillation (NAO) and the Arctic oscillation (AO) can account for the most important climate variability in the Northern Hemisphere. In fact, it has been found that both atmospheric oscillations affect the SST in the Alboran Sea (*Báez et al., 2013*), but the link between occurrence or productivity of microalgae or seaweeds and atmospheric oscillations remains almost unexplored (*Moore et al., 2008*; *Folland et al., 2009*; *Gamboa et al., 2010*; *Smale et al., 2013*; *Báez et al., 2014*).

The population of *F. guiryi* occurring at the limit of the species' distribution, geographically isolated and composed of a low number of individuals, could experience higher stress than the populations inhabiting the Strait of Gibraltar. This conjecture is based on the notion that peripheral populations of organisms are typical cases of "living at the edge" (*Channell & Lomolino, 2000*; *Helmuth et al., 2006*; *Eckert, Samis & Lougheed, 2008*; *Peterman et al., 2013*), with organisms showing signs of physiological stress (*Shumaker & Babble, 1980*). In fact, several studies have revealed physiological stress and changes in growth and reproductive success in *F. guiryi* inhabiting the edge of its geographical distribution, compared to core populations (*Ferreira et al., 2014*; *Zardi et al., 2015*). Following this idea, a second aim of this study was to carry out a comparative study of individuals at Tarifa (Strait of Gibraltar) (Figs. 1A–1C) vs. Punta Calaburras, during the summer of 2011, a year when thalli occurred at both places. For this purpose, we evaluated stress via a physiological approach based on photosynthesis as well as the whole-organism response by assessing developmental stability. Quantum yield from photosystem II photochemistry responds to the alteration of optimum conditions, which could indicate that photosynthesis is being affected (*Baker & Oxborough, 2004*). On the other hand, individuals integrate stress conditions throughout their ontogeny, termed developmental instability (*Clarke & McKenzie, 1987*; *McKenzie & Clarke, 1988*; *Emlen, Freeman & Graham, 1993*; *Palmer, 1994*). Developmental instability is more sensitive than traditional measures of stress (*Graham, Freeman & Emlen, 1993*; *Clarke, 1995*). It is responsive to a wide range of stressors (*Zakharov, 1992*; *Graham, Freeman & Emlen, 1993*) and it is ideally suited for detecting stress in the field (*Graham, Freeman & Emlen, 1993*). Therefore, it could be expected that a higher rate of development instability arises at the limits of distribution of organisms, compared to instability in core populations, because in peripheral areas combinations of environmental factors occur that adversely affect growth, reproduction, and ultimately survival of organisms (*Zakharov, 1992*; *Clarke, 1995*).

## MATERIALS AND METHODS

### Sampling locations and surveys

Study sites were Punta Calaburras (36°30′28″N, 004°38′08″W) and Tarifa (36°00′03″N, 005°36′37″W) (Fig. 1A). Both locations have a similar hot-summer Mediterranean climate (group Csa, Köppen-Geiger climate classification system). Overall mean precipitation ranges from 0 or 1 mm in July, to 93 mm or 146 mm in November, at Punta Calaburras or Tarifa, respectively. Air temperature is also similar at both study sites (overall minimum mean in January of 12.5 °C in Punta Calaburras, and of 11.8 °C in Tarifa; overall maximum mean in August of 25.2 °C in Punta Calaburras vs. 23.4 °C in Tarifa). Surface water

temperature ranges from 15–16 °C in March to 22–24 °C in August in both locations. Moreover, although environmental conditions at both sampling points differ at short-term, a biogeographical subregion (from the Strait of Gibraltar to Punta Calaburras) has been recognized based on the oceanographic conditions in the northern Mediterranean-Atlantic transition zone. This subregion shows a close agreement with the species composition of the littoral and sublittoral benthic communities (*Bermejo et al., 2015*). Consequently, we assumed that comparison of both populations of *F. guiryi* is worthwhile.

The sampling location at Punta Calaburras consists of rocks (schists and gneiss) protruding from the mean sea level up to 30 cm at low spring tide, occupying a surface area of ca. 150 m². The thalli, when they are present, proliferate over almost all the rocks. Consequently, all emergent rocks were visually surveyed during low spring tides, during ca. 1 h searches across all microhabitats. The survey allowed us to detect macrothalli but not possible resilient, microscopic stages. The presence of thalli of *F. guiryi* was checked annually in February–March, and in July-August (Table 1), from 1990 to 2015. Tarifa presents a subhorizontal sandstone platform where *F. guiryi* proliferates. The location was studied in midsummer 2011. To provide consistency in the comparisons with the population at Punta Calaburras, thalli of the alga growing up to 30 cm above low spring tide level were collected and analyzed for developmental instability and photosynthetic performance (see below).

We observed thalli of *F. guiryi* at Punta Calaburras every year of the survey in February–March, but thalli did not always persist into midsummer. The time series of the occurrence of thalli in midsummer was initially analysed taking into account two aspects: the distribution of the presence of thalli and the tendency of the occurrence. The annual occurrence distribution was checked with the test due to *Prahl (1999)* for a stationary Poisson process (randomly distributed throughout the time). The tendency of the annual occurrence distribution was analysed by the Laplace test (*Cox & Lewis, 1978*).

## Analysis of the annual occurrence as a function of oceanographic and atmospheric variables

The relationship between the presence/absence in midsummer of *F. guiryi* thalli at Punta Calaburras from 1990 to 2015, and the temperature, rainfall, NAO, AO and SST, was addressed by binary logistic regressions, widely used for establishing relationships between environmental independent variables and the probability response of target variables (*Zuur, Ieno & Smith, 2007*). Temperature and rainfall mean monthly data were obtained from the Agencia Española de Meteorología (Fuengirola station, 4 km to the east of Punta Calaburras). Of the atmospheric oscillations, NAO is the most important mechanism responsible for the interannual climate variability in SW Europe, particularly during the winter (*Hurrell, 1995*; *Hurrell et al., 2003*). The AO also affects the overall mean of weather conditions in SW Europe. According to *Thompson & Wallace (1998)* the AO explains anomalies in the Arctic region according to the polar vortex. Thus, when the AO index is positive (characterized by a strengthening of the polar vortex), surface pressure is low in the polar region, and the opposite occurs when the index is negative. Monthly AO and NAO index values were obtained from

**Table 1** Time series (1990–2015) of presence (1)/ absence (0) of *Fucus guiryi* at Punta de Calaburras, and overall mean values of NAO and AO indexes (dimensionless), sea surface temperature (SST; units in °C), air temperature (AT; units in °C) and rainfall (R; units in mm), for the previous month or the overall mean for the six previous months (suffixes 1–6).

| Year | Fucus | $AO_1$ | $AO_2$ | $AO_3$ | $AO_4$ | $AO_5$ | $AO_6$ | $NAO_1$ | $NAO_2$ | $NAO_3$ | $NAO_4$ | $NAO_5$ | $NAO_6$ | $SST_1$ | $SST_2$ | $SST_3$ | $SST_4$ | $SST_5$ | $SST_6$ | $AT_1$ | $AT_2$ | $AT_3$ | $AT_4$ | $AT_5$ | $AT_6$ | $R_1$ | $R_2$ | $R_3$ | $R_4$ | $R_5$ | $R_6$ |
|---|---|---|---|---|---|---|---|---|---|---|---|---|---|---|---|---|---|---|---|---|---|---|---|---|---|---|---|---|---|---|---|
| 1990 | 1 | 0.30 | 0.62 | 1.04 | 1.53 | 1.90 | 1.75 | −0.02 | −0.78 | 0.15 | 0.48 | 0.66 | 0.73 | 16.2 | 16.7 | 16.2 | 16.0 | 15.7 | 15.7 | 21.4 | 20.3 | 18.8 | 17.9 | 17.3 | 16.6 | 25 | 153 | 118 | 324 | 346 | 469 |
| 1991 | 1 | −0.12 | 0.19 | 0.30 | 0.09 | −0.10 | 0.04 | −0.82 | −0.37 | −0.15 | −0.16 | 0.08 | 0.21 | 17.5 | 16.9 | 16.1 | 15.6 | 15.1 | 14.9 | 21.9 | 20.0 | 18.4 | 17.4 | 16.3 | 15.8 | 0 | 371 | 267 | 325 | 307 | 377 |
| 1992 | 1 | −0.30 | 0.52 | 0.17 | 0.38 | 0.52 | 0.53 | 0.20 | 1.42 | 1.56 | 1.39 | 1.33 | 1.08 | 14.4 | 15.8 | 15.3 | 15.2 | 14.9 | 14.7 | 18.8 | 19.0 | 18.2 | 17.3 | 16.4 | 15.7 | 270 | 140 | 493 | 472 | 497 | 450 |
| 1993 | 1 | −0.52 | −1.06 | −0.85 | −0.45 | −0.32 | 0.31 | −0.59 | −0.69 | −0.13 | 0.07 | 0.15 | 0.40 | 16.2 | 15.6 | 15.2 | 15.0 | 14.9 | 14.7 | 20.7 | 19.2 | 17.9 | 16.8 | 16.0 | 15.4 | 0 | 103 | 209 | 323 | 384 | 400 |
| 1994 | 1 | 1.61 | 0.75 | 0.57 | 0.90 | 0.55 | 0.41 | 1.52 | 0.48 | 0.70 | 0.84 | 0.76 | 0.81 | 17.6 | 16.2 | 15.6 | 15.4 | 15.0 | 14.8 | 20.8 | 19.4 | 18.2 | 17.5 | 16.6 | 16.0 | 25 | 208 | 397 | 482 | 599 | 579 |
| 1995 | 0 | −0.11 | −0.50 | −0.66 | −0.39 | −0.03 | −0.05 | 0.13 | −0.68 | −0.74 | −0.24 | 0.04 | 0.19 | 17.7 | 17.1 | 16.5 | 16.1 | 15.7 | 15.4 | 20.6 | 19.9 | 18.6 | 17.8 | 17.0 | 16.3 | 0 | 19 | 13 | 34 | 143 | 368 |
| 1996 | 0 | 0.50 | 0.14 | −0.42 | −0.68 | −0.51 | −0.63 | 0.56 | −0.25 | −0.22 | −0.23 | −0.20 | −0.18 | 19.0 | 17.3 | 16.8 | 16.3 | 15.9 | 15.7 | 21.6 | 19.7 | 18.7 | 17.7 | 16.7 | 16.4 | 65 | 194 | 287 | 284 | 378 | 334 |
| 1997 | 0 | −0.81 | −0.89 | −0.48 | −0.09 | 0.31 | 0.18 | −1.47 | −0.88 | −0.92 | −0.33 | 0.08 | −0.02 | 15.7 | 15.7 | 16.0 | 16.1 | 15.9 | 15.6 | 20.7 | 19.9 | 19.2 | 18.4 | 17.7 | 16.9 | 0 | 3 | 119 | 303 | 502 | 797 |
| 1998 | 0 | −0.71 | −0.14 | −0.11 | −0.14 | −0.15 | −0.47 | −2.72 | −2.02 | −1.57 | −0.96 | −0.79 | −0.60 | 17.7 | 16.8 | 15.9 | 15.8 | 15.6 | 15.4 | 21.1 | 19.2 | 18.3 | 17.6 | 16.9 | 16.4 | 45 | 315 | 252 | 436 | 531 | 550 |
| 1999 | 1 | 0.71 | 0.47 | 0.41 | −0.07 | 0.04 | 0.05 | 1.12 | 1.02 | 0.36 | 0.33 | 0.32 | 0.40 | 18.5 | 17.0 | 16.3 | 16.0 | 15.6 | 15.4 | 21.8 | 20.3 | 19.2 | 18.1 | 17.0 | 16.3 | 145 | 450 | 358 | 346 | 277 | 234 |
| 2000 | 1 | 0.59 | 0.78 | 0.43 | 0.21 | 0.38 | 0.53 | −0.03 | 0.78 | 0.51 | 0.57 | 0.80 | 0.77 | 20.4 | 18.5 | 16.9 | 16.6 | 16.3 | 15.9 | 22.6 | 20.7 | 19.0 | 18.1 | 17.4 | 16.5 | 147 | 472 | 522 | 472 | 729 | 766 |
| 2001 | 1 | −0.02 | 0.22 | 0.45 | −0.09 | −0.19 | −0.32 | −0.20 | −0.11 | −0.07 | −0.37 | −0.21 | −0.13 | 18.2 | 17.2 | 16.7 | 16.3 | 15.9 | 15.5 | 22.6 | 20.4 | 19.3 | 18.6 | 17.6 | 17.0 | 9 | 155 | 137 | 226 | 181 | 175 |
| 2002 | 1 | 0.57 | 0.49 | 0.57 | 0.66 | 0.79 | 0.88 | 0.38 | 0.08 | 0.45 | 0.51 | 0.63 | 0.60 | 16.6 | 16.3 | 16.1 | 15.8 | 15.7 | 15.5 | 21.3 | 20.1 | 18.7 | 17.8 | 17.1 | 16.5 | 916 | 600 | 909 | 980 | 785 | 773 |
| 2003 | 1 | −0.10 | 0.46 | 0.25 | 0.42 | 0.36 | 0.22 | −0.07 | −0.03 | −0.08 | 0.02 | 0.14 | 0.14 | 19.0 | 18.4 | 17.5 | 17.0 | 16.5 | 16.1 | 22.9 | 21.5 | 19.8 | 18.6 | 17.5 | 16.7 | 147 | 366 | 253 | 498 | 725 | 651 |
| 2004 | 1 | −0.24 | −0.17 | −0.25 | −0.11 | −0.39 | −0.61 | −0.89 | −0.35 | 0.15 | 0.37 | 0.27 | 0.17 | 19.1 | 17.5 | 16.8 | 16.4 | 16.0 | 15.8 | 22.5 | 19.9 | 18.6 | 17.6 | 16.7 | 16.3 | 3 | 246 | 386 | 488 | 430 | 462 |
| 2005 | 0 | −0.38 | −0.57 | −0.40 | −0.64 | −0.76 | −0.58 | −0.05 | −0.65 | −0.53 | −0.86 | −0.70 | −0.33 | 18.3 | 17.5 | 16.7 | 16.1 | 15.6 | 15.4 | 22.2 | 20.9 | 19.6 | 18.2 | 16.9 | 16.0 | 0 | 56 | 46 | 229 | 560 | 487 |
| 2006 | 1 | 1.07 | 0.61 | 0.45 | −0.06 | −0.08 | −0.09 | 0.84 | −0.15 | 0.31 | −0.09 | −0.17 | 0.07 | 18.7 | 18.3 | 17.3 | 16.4 | 15.9 | 15.7 | 21.6 | 20.8 | 19.6 | 18.6 | 17.7 | 17.3 | 103 | 233 | 450 | 736 | 791 | 816 |
| 2007 | 0 | −0.55 | 0.17 | 0.29 | 0.52 | 0.15 | 0.47 | −1.31 | −0.32 | −0.16 | 0.24 | 0.10 | 0.12 | 16.0 | 15.7 | 15.3 | 15.0 | 15.0 | 15.1 | 20.6 | 19.8 | 18.5 | 17.6 | 17.0 | 16.3 | 1,283 | 1,640 | 1,110 | 1,365 | 1,293 | 1,219 |
| 2008 | 0 | −0.09 | −0.65 | −0.58 | −0.29 | −0.05 | 0.10 | −1.39 | −1.56 | −1.39 | −1.03 | −0.67 | −0.41 | 18.4 | 17.2 | 16.7 | 16.5 | 16.2 | 16.1 | 21.7 | 20.0 | 19.0 | 18.2 | 17.5 | 17.0 | 673 | 2,209 | 3,031 | 2,482 | 2,132 | 1,777 |
| 2009 | 0 | −1.35 | −0.08 | 0.27 | 0.23 | 0.05 | 0.18 | −1.21 | 0.24 | 0.09 | 0.21 | 0.18 | 0.15 | 18.2 | 17.2 | 16.4 | 16.0 | 15.6 | 15.3 | 21.8 | 20.9 | 19.3 | 18.2 | 17.2 | 16.5 | 198 | 474 | 383 | 451 | 373 | 311 |
| 2010 | 0 | −0.01 | −0.47 | −0.40 | −0.41 | −1.18 | −1.42 | −0.82 | −1.15 | −1.01 | −0.98 | −1.18 | −1.17 | 17.9 | 17.2 | 17.0 | 16.4 | 16.0 | 15.7 | 20.6 | 19.4 | 18.5 | 17.5 | 16.9 | 16.3 | 246 | 466 | 370 | 602 | 547 | 456 |
| 2011 | 1 | −0.86 | −0.45 | 0.46 | 0.70 | 0.88 | 0.45 | −1.28 | −0.67 | 0.38 | 0.44 | 0.49 | 0.26 | 19.2 | 18.6 | 17.8 | 17.2 | 16.9 | 16.7 | 21.5 | 20.6 | 19.5 | 18.4 | 17.5 | 16.8 | 588 | 761 | 560 | 809 | 700 | 583 |
| 2012 | 0 | −0.67 | −0.25 | −0.18 | 0.12 | 0.09 | 0.04 | −2.53 | −1.72 | −0.99 | −0.43 | −0.26 | −0.02 | 18.4 | 17.3 | 16.3 | 15.8 | 15.6 | 15.5 | 22.1 | 20.5 | 19.1 | 17.9 | 16.6 | 16.0 | 32 | 899 | 1,047 | 804 | 643 | 536 |
| 2013 | 0 | 0.55 | 0.52 | 0.45 | −0.46 | −0.57 | −0.57 | 0.52 | 0.54 | 0.59 | 0.04 | −0.06 | 0.01 | 17.1 | 16.3 | 15.9 | 15.4 | 15.0 | 14.9 | 20.3 | 19.1 | 18.0 | 17.2 | 16.4 | 16.0 | 8 | 294 | 361 | 271 | 217 | 181 |
| 2014 | 0 | −0.51 | −0.02 | 0.31 | 0.53 | 0.44 | 0.20 | −0.97 | −0.95 | −0.53 | −0.20 | 0.11 | 0.14 | 18.1 | 18.1 | 17.7 | 17.0 | 16.4 | 16.1 | 21.3 | 20.1 | 19.2 | 18.1 | 17.3 | 16.8 | 691 | 346 | 235 | 177 | 185 | 189 |
| 2015 | 0 | 0.43 | 0.59 | 0.80 | 1.06 | 1.06 | 1.06 | −0.07 | 0.04 | 0.27 | 0.56 | 0.72 | 0.89 | 17.7 | 17.1 | 16.4 | 16.0 | 15.7 | 15.5 | 21.7 | 20.1 | 18.6 | 17.7 | 16.8 | 16.2 | 1,175 | 1,008 | 744 | 607 | 486 | 405 |

the free-access web sites of the US National Oceanic and Atmospheric Administration, http://www.cpc.ncep.noaa.gov/products/precip/CWlink/daily_ao_index/ao.shtml and http://www.cpc.ncep.noaa.gov/products/precip/CWlink/pna/nao.shtml, respectively. Finally, the power of the Atlantic current entering the Mediterranean Sea was estimated by the SST values close to Punta Calaburras (the higher the flow of AJ water, the lower SST is; *Parrilla & Kinder, 1987*). Data of SST were obtained from the free-access web site from the Centro Oceanográfico de Málaga (sede Fuengirola), Instituto Español de Oceanografía, http://www.ma.ieo.es/gcc/playafuengirola_taireyagua_anomalias.xls.

For the analysis, we tested the monthly values of environmental variables from the same month as well as the overall mean figures from two to six previous months. We assessed the significance of the variables in the model using the Wald test (*Wald, 1943*), the calibration of the model using the Hosmer & Lemeshow test (*Hosmer & Lemeshow, 1980*), its discrimination capacity using the area under the curve (AUC) of the receiving operator characteristics (*Lobo, Jiménez-Valverde & Real, 2008*), and its explanatory power using the Nagelkerke $R^2$ (*Nagelkerke, 1991*).

Additionally, we used relevant probability levels to assess the environmental conditions that favoured the presence of *F. guiryi* thalli, the opening gap between the values considered as clearly probable ($p > 0.6$) or clearly improbable ($p < 0.4$). It must be taken into account that $p = 0.5$ means that the presence or the absence of the thalli have a similar probability. We then compared the correct classification rate of the models for years clearly probable and clearly improbable for a presence of thalli, and simultaneously we identified the levels of the environmental variables associated with the relevant probability levels.

**Developmental instability**

All the species in the genus *Fucus* exhibit self-symmetry, i.e., symmetry across scale (*Corbit & Garbary, 1995*). We estimated the developmental instability in *F. guiryi* individuals by deviations of the self-symmetry of thalli, by using the box-counting procedure (*Mandelbrot, 1983*; *Iannaccone & Khokha, 1996*). At Punta Calaburras, a single thallus was collected from the center of a randomly placed $15 \times 15$ cm quadrat, from each of the 20 different protruding rocks (those reaching the maximum height over the sea surface). At Tarifa, a 5 m length horizontal transect was located parallel to the sea surface and a thallus was collected every 20 cm from the center of a quadrat of $15 \times 15$ cm. At both study sites, the sampling stopped when the twenty-first thallus lacking grazing marks was collected. This precaution was taken because grazing marks alter the thallus shape and, consequently, the self-symmetry; thus, individuals showing no damage were selected (around 15% of thalli showed herbivore marks, especially by the fish *Sarpa salpa*). Curiously, grazing marks were exclusively found on thalli collected in Tarifa, possibly due to the fact that the thalli inhabiting Punta Calaburras are located in the upper part of the tidal range of the area. Thalli were placed between two transparent acetate foils, avoiding overlapping the fronds, and they were scanned in TIFF format (300 ppi). The scanned images were superimposed on grids with exponentially increasing box sizes (0.125, 0.25, 0.5, 1, 2 and 4 cm$^2$). The number of boxes in which at least part of the thallus occurred were counted using an image analysis system Visilog 6.3 (Noesis, France). Twenty independent thalli were processed

from each location. Because the overall positioning of the boxes can affect the results of a box count (*Walsh & Watterson, 1993*; *Schulze, Hutchings & Simpson, 2008*), the counting of the boxes was carried out three times (named "replicates"), repositioning the thalli over the acetate foils with the grids every time. We then regressed the natural log of the number of occupied boxes against the natural log of the size of each box. The absolute value of the slope of the regression line is the fractal dimension (a measure of the space filled by the individual). Developmental instability is the degree to which the individual failed to fit the idealized phenotype, and is measured as the standard error of the estimate ($S_{Y \cdot X}$, computed as the square root of the residual mean square of the ANOVA regression of the linear fit). The value of $S_{Y \cdot X}$ is an overall indication of the accuracy with which the fitted regression function predicts the dependence of $Y$ on $X$. Under non-stressful conditions, all points should lie on the regression line.

A two-level nested ANOVA (model: $y$ = overall mean + locations + replicates [locations] + error) was performed to compare the $S_{Y \cdot X}$ values. The factor "locations" correspond to the Tarifa and Punta Calaburras populations, whereas the factor "replicates [locations]" corresponds to the three independent measurements of $S_{Y \cdot X}$ of each thallus from both locations. The homogeneity of variances was previously checked with Bartlett's test.

## Measurement of natural solar radiation and temperature in air and water

The measurements were carried out on 16th July 2011 at Tarifa, and on the following day at Punta Calaburras. Daily changes in PAR ($\lambda = 400$–$700$ nm) were measured using a LI-190R PAR sensor connected to a LI-1400 data logger (LI-COR, Lincoln, NE, USA). The ultraviolet A ($\lambda = 315$–$400$ nm) and ultraviolet B ($\lambda = 280$–$315$ nm) bands were measured using a RM12 device (Dr. Gröbel, Ettlingen, Germany) connected to the respective UVA and UVA sensors. Measurements were made every 30 min, and data were fit to a single sinusoid with the free software PAST ver. 2.17 (*Hammer, Harper & Ryan, 2001*); the daily doses of each channel were calculated by integrating the area under the sinusoidal curves. Air temperature ($\pm 0.1\ ^\circ$C) in the shade was measured with a sensor connected to a LI-1400 data logger. Seawater temperature ($\pm 0.1\ ^\circ$C) was measured with the temperature sensor of a Crison2 OXI-92 (Crison, Barcelona, Spain) oxymeter.

### *In vivo* measurements of chlorophyll *a* fluorescence

A day-long record (from 06.00 to 18.00, Coordinated Universal Time –UTC-) of the photosynthetic performance of *F. guiryi* was carried out on the same days as the solar radiation measurements. The sampling day was selected to correspond to a spring tide, with a maximum tidal height ca. 1.2 m at Tarifa and ca. 0.4 m at Punta Calaburras (Fig. 2). Because the weather was sunny without clouds and with similar air and seawater temperatures (see 'Results'), we assumed that the photosynthetic performance measured on the two consecutive days at the different locations should be comparable. Five independent thalli were randomly collected before sunset from the higher eulittoral of Tarifa and Punta Calaburras. For this purpose, we initially randomly collected 20 thalli at each location, as explained in the *Developmental instability* section above. Then, the thalli were

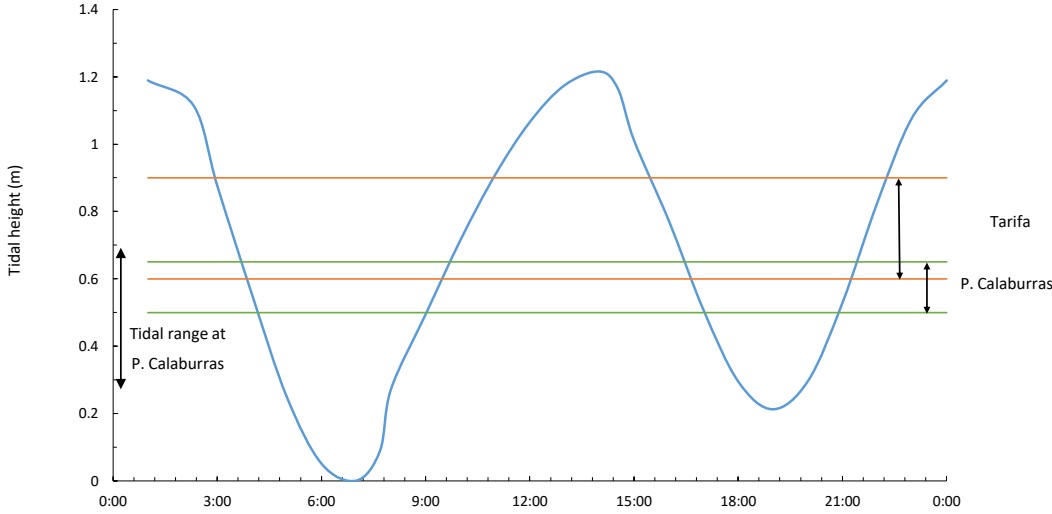

**Figure 2  Tidal range and vertical ranges of *Fucus guiryi* at Tarifa and at Punta Calaburras.** Daily (hours:minutes, Coordinated Universal Time) tidal height at Tarifa and maximum tidal range at Punta Calaburras on 16th July 2011, and vertical ranges of *Fucus guiryi* at Tarifa and Punta Calaburras. Daily tidal regime at Punta Calaburras has not been included because no data are available.

randomly numbered from 1 to 20, and we selected five thalli whose numbers correspond to those that had been generated with a random integer generator accessible at a free-access web site: https://www.random.org/integers/. The thalli were placed in a 25 L white polyvinylchloride tank in natural seawater from the collection site. The tank was placed close to the attachment site of the alga, in an unshaded place. To avoid nutrient depletion and changes in temperature in the tank, seawater was renewed completely every 10 min. The measurements were carried out every 2 h from sunrise to sunset; from 10.00 to 14.00 UTC; an extra set of measurements was carried out on thalli exposed to air from 09.30 UTC.

Fresh mass (FM) on thalli *in situ* was measured with an ELB120 portable analytical balance (±0.01 g) (Shimadzu, Kyoto, Japan) after thalli were blotted dry with paper towel. The same samples were transported to the laboratory and dried at 60 °C for 48 h to determine their dry mass (DM). The percentage of water content of thalli was determined as:

$$\% \text{ Water content} = \left(\frac{\text{FM} - \text{DM}}{\text{FM}}\right) \times 100.$$

Chlorophyll fluorescence was measured using a portable pulse amplitude modulated PAM-2000 fluorimeter (Walz, Effeltrich, Germany) following *Schreiber, Schliwa & Bilger (1986)*. The optimal or potential quantum efficiency ($F_v/F_m$) was measured in thalli exposed to darkness for 30 min.

The relative electron transport rate ($ETR_{rel}$) was estimated as:

$$ETR_{rel} = \Phi_{PSII} \times I$$

where $I$ is the incident irradiance of PAR ($\lambda = 400{-}700$ nm) and $\Phi_{PSII}$ is the quantum yield of PSII photochemistry.
The contribution of the location and the time of day, on water content of thalli (in air), and $F_v/F_m$, $\Phi_{PSII}$ and $ETR_{rel}$ (in air and water), was analysed by a two-way, model I ANOVA. Differences, when obtained, were checked by the Student-Newman Keuls (SNK) procedure. The homogeneity of variances was previously checked with the Bartlett's test. The Pearson's correlation coefficient was computed for the relationships between hydration of thalli and photosynthetic performance parameters.

## Statistical software analysis

The exp and tendency tests in the time series were performed using the free software PAST ver. 2.17 (*Hammer, Harper & Ryan, 2001*) accessible at http://nhm2.uio.no/norlex/past/download.html. The remaining statistical analyses were carried out using *R Core Team (2013)*.

# RESULTS

## Analysis of the time series of occurrence

The occurrences in midsummer of thalli of *F. guiryi* at Punta Calaburras through the years 1990 to 2015 were clustered ($M = 0.96$; $M$-expected $= 0.36$, $p < 0.0001$) from 1990–1994 and 1999–2004, with sporadic occurrences in 2006 and 2011 (Table 1). A trend in the occurrence of thalli in midsummer throughout of the time series was not detected ($U$-Laplace test $= -3.8 \times 10^{-15}$, $p = 1$).

We found a significant positive relationship between the NAO for the months from April to June ($NAO_3$) of each year and the probability of the presence of *F. guiryi* ($\chi^2 = 13.530$, $df = 1$, $p = 0.0002$; Wald's test $= 5.994$, $df = 1$, $p = 0.014$; Table 2) according to the logit $y$ function:

$$y = 3.418 \times NAO_3 + 0.239.$$

The 95% confidence limits for the intercept and for the explanatory variable $NAO_3$ were $[-0.779, 1.263]$ and $[0.682, 6.138]$, respectively; that is to say, a logit $y$ function in which the intercept is deleted could also be used because its contribution to the model was not significant (the confidence interval includes the figure 0). This model was well calibrated (Hosmer and Lemeshow's test $= 4.145$, $df = 7$, $p = 0.7661$), meaning that the differences between observed and predicted frequencies were not significant. The overall ability of discrimination and the general explanatory power of the model were high (AUC $= 0.876$ and Nagelkerke $R^2 = 0.541$, respectively).

According to the logit $y$ function, the probability $p$ of the occurrence of *F. guiryi* thalli in midsummer computed as $\exp^y/(1 + \exp^y)$, was calculated (Fig. 3). Based on relevant $p$ values, we estimated the correct classification of years in which the $NAO_3$ index favoured the presence or the absence of *F. guiryi* thalli. The model clearly identified three of four highly probable years ($p > 0.6$, corresponding to $NAO_3 > 0.048$) for the presence of *F. guiryi* thalli, and simultaneously, all of the clearly improbable years ($p > 0.4$, corresponding to $NAO_3 < -0.189$) were correctly assigned.
**Table 2** **Binary logistic regression between presence/absence of *Fucus guiryi* as a function of NAO and AO indexes, sea surface temperature (SST), air temperature (AT) and rainfall (R).** Value of $\chi^2$-test ($df = 1$ in all of the cases) and associated probability, and Akaike Information Criterion (AIC), in the first step of the binary logistic regression between presence/absence of *Fucus guiryi* as a function of NAO and AO indexes, sea surface temperature (SST), air temperature (AT) and rainfall (R), for the previous month or the overall mean for the six previous months (suffixes 1–6). The asterisk shows the independent variable selected for the analyses based on the higher explanatory power.

| Variables | $\chi^2$ | $p$ | AIC |
|---|---|---|---|
| $AO_1$ | 3.785 | 0.045 | 36.035 |
| $AO_2$ | 4.735 | 0.029 | 35.308 |
| $AO_3$ | 4.656 | 0.031 | 35.387 |
| $AO_4$ | 3.208 | 0.073 | 36.835 |
| $AO_5$ | 3.364 | 0.066 | 36.680 |
| $AO_6$ | 3.384 | 0.060 | 36.516 |
| $NAO_1$ | 5.532 | 0.018 | 34.511 |
| $NAO_2$ | 6.824 | 0.008 | 33.220 |
| $NAO_3^*$ | 13.530 | 0.0002 | 26.513 |
| $NAO_4$ | 10.543 | 0.001 | 29.501 |
| $NAO_5$ | 9.910 | 0.001 | 30.133 |
| $NAO_6$ | 9.431 | 0.002 | 30.612 |
| $SST_1$ | 0.028 | 0.866 | 40.015 |
| $SST_2$ | 0.386 | 0.543 | 39.658 |
| $SST_3$ | 0.006 | 0.941 | 40.038 |
| $SST_4$ | 0.008 | 0.930 | 40.036 |
| $SST_5$ | 0.011 | 0.916 | 40.033 |
| $SST_6$ | 0.019 | 0.890 | 40.024 |
| $AT_1$ | 0.899 | 0.343 | 39.168 |
| $AT_2$ | 0.760 | 0.383 | 39.294 |
| $AT_3$ | 0.670 | 0.769 | 39.991 |
| $AT_4$ | 0.480 | 0.826 | 39.967 |
| $AT_5$ | 0.017 | 0.896 | 40.036 |
| $AT_6$ | 0.005 | 0.942 | 40.035 |
| $R_1$ | 1.184 | 0.277 | 38.860 |
| $R_2$ | 2.378 | 0.123 | 37.665 |
| $R_3$ | 1.110 | 0.292 | 38.933 |
| $R_4$ | 0.433 | 0.511 | 39.611 |
| $R_5$ | 0.386 | 0.534 | 39.658 |
| $R_6$ | 0.263 | 0.608 | 39.781 |

## Developmental instability

The standard error of the regression ($S_{Y \cdot X}$) derived from the box-counting method was used as a proxy for developmental instability in *F. guiryi*. The comparison of the $S_{Y \cdot X}$ values showed that the replicates [locations] were not significantly different (nested ANOVA; $F = 0.0002$, $df = 4$ and 114, $p = 1.000$) suggesting that the reproducibility of the method was reliable. The $S_{Y \cdot X}$ values ranged from 0.025 to 0.162 (overall mean $= 0.094 \pm 0.038$) in the thalli from Tarifa, and from 0.037 to 0.153 (overall mean $= 0.090 \pm 0.025$) in the algae
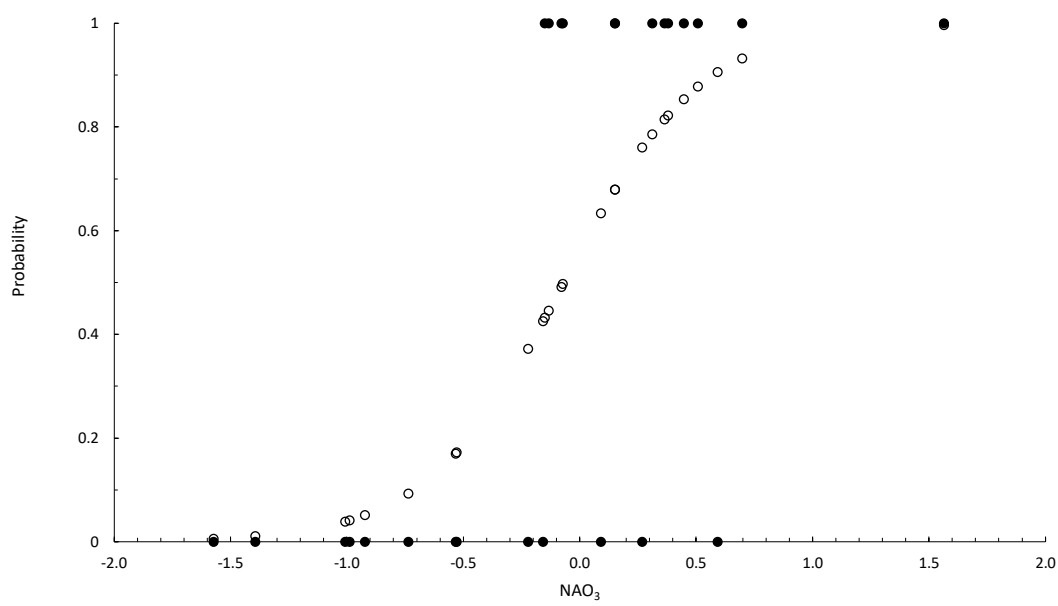

**Figure 3** **Binary logistic regression of occurence of *Fucus guiryi* as a function of NAO₃.** Probability (white circles) and presence (1)/absence (0) (black circles) of *Fucus guiryi* at Punta de Calaburras from 1990 to 2015, as a function of the North Atlantic Oscillation averaged for the months between April and June (NAO₃), as an explanatory independent environmental variable.

collected at Punta Calaburras, being significantly higher, at an α level of 0.05, at Tarifa than at Punta Calaburras (nested ANOVA; $F = 14.041$, $df = 1$ and 4, $p = 0.040$).

### *In situ* temperature, solar radiation and photosynthetic performance

At Tarifa, the temperature of the air on 16th July 2011 increased from 18.2 °C in early morning to an overall mean of 27.4 ± 0.3 °C between 12.30 and 14.30 UTC, and then declined throughout the afternoon. The temperature of the seawater did not change significantly throughout the day, with an overall mean value of 19.5 ± 0.1 °C. The air temperature records at Punta Calaburras were 21.1 °C in early morning and a maximum of 28.2 °C at noon; the seawater temperature ranged from 19.1 to 19.8 ° C.

The daily profile of the solar irradiance recorded at Tarifa showed a symmetrical pattern centered at noon, typical for a clear blue sky (Fig. 4). Daily doses of solar radiation were 9228.25, 1109.70 and 13.03 kJ m$^{-2}$ for PAR, ultraviolet A and ultraviolet B, respectively (Fig. 4). Solar radiation data recorded at Punta Calaburras the following day were similar (data not shown), with doses differing <±3%.

The $F_v/F_m$ figures ranged from 0.674 ± 0.035 to 0.732 ± 0.034 during the day in permanently submerged thalli (Fig. 5A). The $F_v/F_m$ values were similar at both locations but a highly significant effect of time of day was detected (Table 3). The interaction between sampling location and time of day was not significant (Table 3). Under simulated emerged conditions, the values of $F_v/F_m$ significantly decreased from 10:00 to 14:00 (Fig. 5A), with a greater decrease in thalli from Punta Calaburras than those from Tarifa (Table 3). A significant interaction between locations and time of day was also found (Table 3).
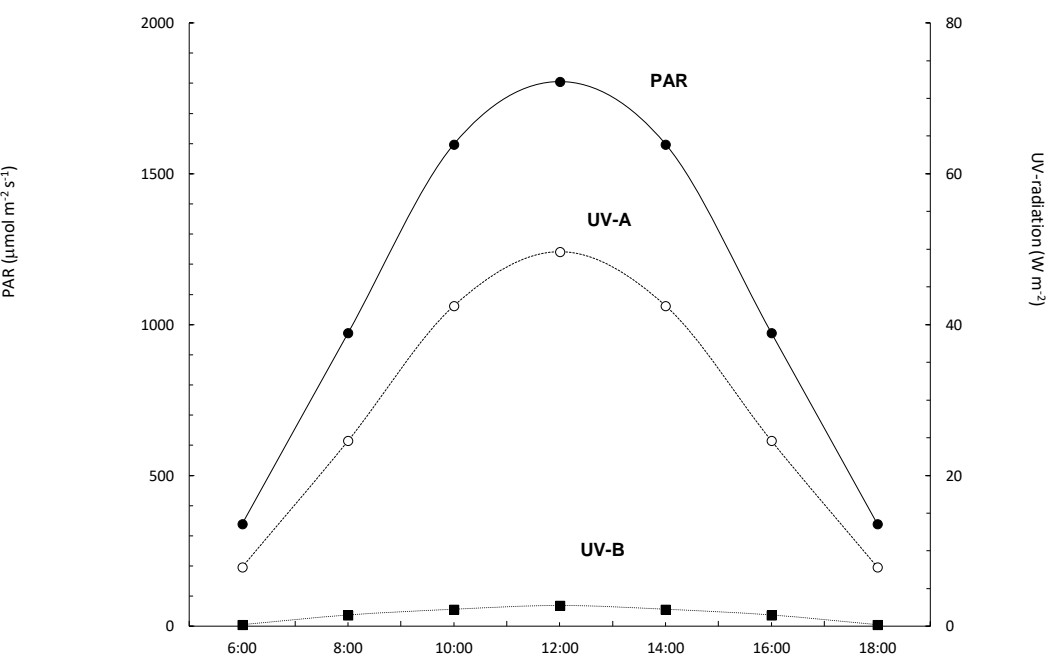

**Figure 4  Daily course of PAR, UV-A and UV-B solar radiation.** Daily course of PAR (black circles, continuous line), and UV-A (white circles, dotted line) and UV-B (black squares, dotted line) radiation at Tarifa, on 16th July 2011.

The $\Phi_{PSII}$ figures ranged from $0.307 \pm 0.023$ to $0.732 \pm 0.028$ during the day in permanently submerged thalli (Fig. 5B). The $\Phi_{PSII}$ values were similar at both locations (Table 3) but a highly significant effect of time of day was detected (Table 3). The interaction between sampling location and time of day was not significant (Table 3). Under simulated emerged conditions the values of $\Phi_{PSII}$ significantly decreased from 10:00 to 14:00 (Fig. 5B; Table 3), with a greater decrease in thalli from Calaburras than those from Tarifa (Fig. 5B; Table 3). A significant interaction between locations and time of day was also found (Table 3).

The $ETR_{rel}$ figures ranged from $175.3 \pm 11.4$ to $671.7 \pm 34.0$ during the day in permanently submerged thalli (Fig. 5C). The $ETR_{rel}$ values were similar at both locations (Table 3) but a highly significant effect of time of day was detected (Table 3). The interaction between sampling location and time of day was not significant (Table 3). Under simulated emerged conditions the values of $ETR_{rel}$ significantly decreased from 10:00 to 14:00 (Fig. 5C), with a greater decrease in thalli from Calaburras than those from Tarifa (Fig. 5C; Table 3). A significant interaction between locations and time of day was also found (Table 3).

The water content in algal fronds decreased when they were exposed to air (Fig. 5D), with a greater decrease in thalli from Calaburras than those from Tarifa (Fig. 5D; Table 3). A significant interaction between location and time of day was also found (Table 3).

We found that the hydration level significantly correlated ($p < 0.0001$, $n = 15$) both with $F_v/F_m$ ($r = 0.9407$) and $\Phi_{PSII}$ ($r = 0.9039$).

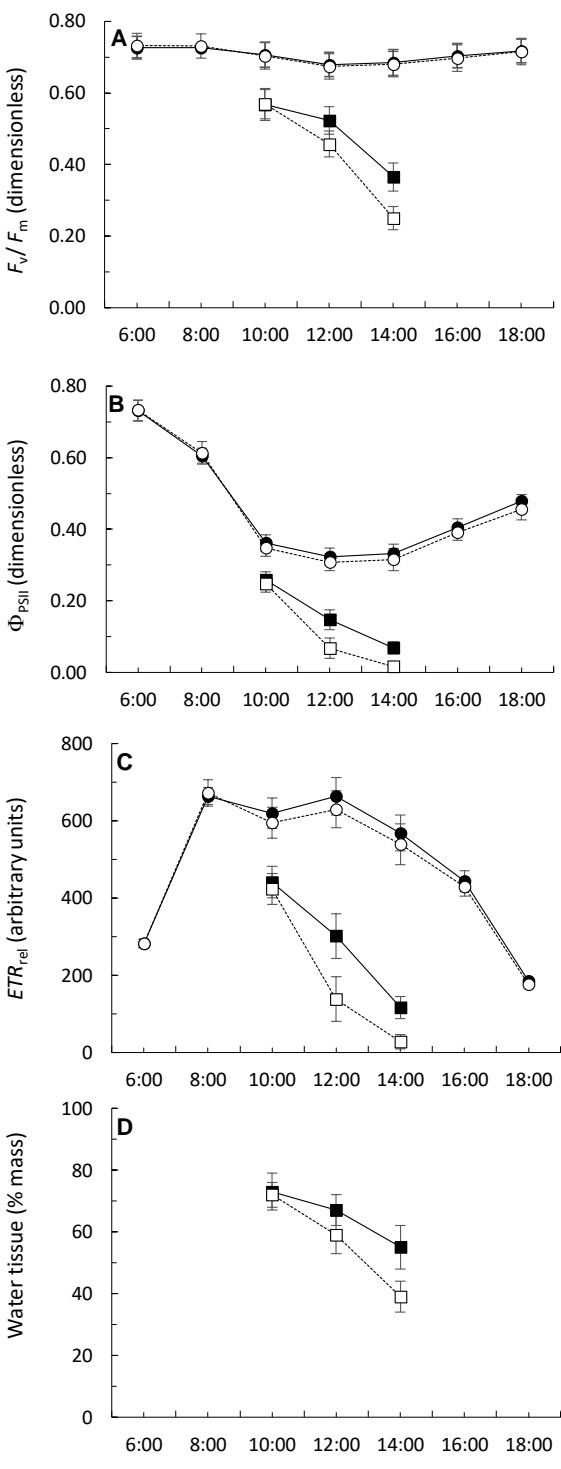

**Figure 5** Daily course (hours:minutes, Coordinated Universal Time) of the $F_v/F_m$ (A), $\Phi_{PSII}$ (B) $ETR_{rel}$ (C) and water tissue content (D) of *Fucus guiryi* from Tarifa (black symbols, continuous lines) and from Punta de Calaburras (white symbols, dotted lines) in air (squares) and water (circles). The measurements were carried out on 16th July 2011 in Tarifa, and the next day in Punta de Calaburras.

**Table 3** Two-way ANOVAS for the comparison of the $F_v/F_m$, $\Phi_{PSII}$, $ETR_{rel}$ and water tissue content of *Fucus guiryi*, and Student-Newman-Keuls (SNK) post-hoc test for significant (95%) sources of variation.

| Variable | Source of variation | df | SS | MS | F | p | SNK |
|---|---|---|---|---|---|---|---|
| $F_v/F_m$ in water | Locations | 1 | 0.0001 | 0.0001 | 2.679 | 0.107 | |
| | Time of day | 6 | 0.0492 | 0.0082 | 296.800 | 0.000 | 6:00 = 8:00 > 18:00 > 10:00 = 16:00 > 14:00 > 12:00 |
| | Locations × Time of day | 6 | 0.0002 | 0.0000 | 1.304 | 0.270 | |
| | Error | 56 | 0.0015 | 0.0000 | | | |
| $F_v/F_m$ in air | Locations | 1 | 0.0269 | 0.0269 | 17.960 | 0.000 | Tarifa > Calaburras |
| | Time of day | 2 | 0.3569 | 0.1785 | 119.000 | 0.000 | 10:00 > 12:00 > 14:00 |
| | Locations × Time of day | 2 | 0.0172 | 0.0086 | 5.726 | 0.009 | |
| | Error | 24 | 0.0360 | 0.0015 | | | |
| $\Phi_{PSII}$ in water | Locations | 1 | 0.0021 | 0.0021 | 3.209 | 0.078 | |
| | Time of day | 6 | 1.5120 | 0.2520 | 384.000 | 0.000 | 6:00 > 8:00 > 18:00 > 16:00 > 10:00 > 14:00 = 12:00 |
| | Locations × Time of day | 6 | 0.0018 | 0.0003 | 0.455 | 0.838 | |
| | Error | 56 | 0.0368 | 0.0007 | | | |
| $\Phi_{PSII}$ in air | Locations | 1 | 0.0167 | 0.0167 | 32.480 | 0.000 | Tarifa >Calaburras |
| | Time of day | 2 | 0.2320 | 0.1160 | 225.500 | 0.000 | 10:00 >12:00 >14:00 |
| | Locations × Time of day | 2 | 0.0061 | 0.0030 | 5.889 | 0.008 | |
| | Error | 24 | 0.0124 | 0.0005 | | | |
| $ETR_{rel}$ in water | Locations | 1 | 3,764 | 3,764 | 3.296 | 0.075 | |
| | Time of day | 6 | 2,154,000 | 358,900 | 314.300 | 0.000 | 8:00 = 12:00 > 10:00 > 14:00 > 16:00 > 6:00 > 18:00 |
| | Locations × Time of day | 6 | 3,617 | 603 | 0.528 | 0.785 | |
| | Error | 56 | 63,940 | 1,142 | | | |
| $ETR_{rel}$ in air | Locations | 1 | 60,450 | 60,450 | 32.800 | 0.000 | Tarifa > Calaburras |
| | Time of day | 2 | 655,700 | 327,800 | 177.900 | 0.000 | 10:00 > 12:00 > 14:00 |
| | Locations × Time of day | 2 | 26,430 | 13,210 | 7.169 | 0.004 | |
| | Error | 24 | 44,230 | 1,843 | | | |
| Water of tissue in air | Locations | 1 | 326.7 | 326.7 | 49.630 | 0.0003 | Tarifa > Calaburras |
| | Time of day | 2 | 3,712.0 | 1,856.0 | 281.900 | 0.0000 | 10:00 > 12:00 > 14:00 |
| | Locations × Time of day | 2 | 175.4 | 87.7 | 13.320 | 0.0001 | |
| | Error | 24 | 158.0 | 6,583.0 | | | |

## DISCUSSION

Populations of fucoids living at the edge of their respective geographical distribution are being studied because shifts have occurred in recent decades (*Viejo et al., 2011*; *Lamela-Silvarrey et al., 2012*; *Fernández, 2016*). A survey of several populations of *F. guiryi* inhabiting the southernmost area of its geographical distribution (from Portugal to Western Sahara) showed that the species is restricted to upwelling areas, which can function as climate change refugia (*Lourenço et al., 2016*). Moreover, these authors highlighted that the population inhabiting the Strait of Gibraltar revealed the greatest genetic differentiation in comparison to other populations. The isolated population of *F. guiryi* inhabiting Punta Calaburras (Alboran Sea, Western Mediterranean Sea) serves as a model to understand

the role of oceanographic and atmospheric conditions on the annual occurrence of thalli in midsummer. Recruitment of few-celled stages could occur from the rapid settlement of nearby parental thalli (*Serrão et al., 1996*) when thalli survive year-to-year. However, after the disappearance of thalli in midsummer of a given year, the re-appearance of *F. guiryi* at Punta Calaburras could be achieved by the presence of cryptic, microscopic stages that survive unfavourable summer months before developing again during cooler months, or after catastrophic events (*Creed, Norton & Harding, 1996*; *Carney & Edwards, 2006*; *Schiel & Foster, 2006*). Thus, local microscopic stages (recruits and holdfast remnants) at Punta Calaburras could function as a "seed bank." An alternative hypothesis postulates that the re-appearance of thalli in winter could result from the recruitment and establishment of embryos transported from neighbouring populations in the Strait of Gibraltar (*Conde & Seoane-Camba, 1982*; *Conde, 1989*), around 80 km from Punta Calaburras (see Fig. 1A). However, *Neiva et al. (2014)* showed that microscopic stages of *Fucus ceranoides* L. are poorly dispersed at scales of kilometres, and this is also possibly true for *F. guiryi*. With our data, we cannot discern whether the recurring re-appearance of thalli in winter is due to immigration or local recruitment, or both. Moreover, we found no regularities nor tendencies in the occurrence along the time series but, obviously, the "switch-on switch-off" pattern of the thalli of *F. guiryi* at Punta Calaburras must be under environmental control. For this reason, we evaluated oceanographic and atmospheric factors that could be involved in the persistence of macroscopic thalli during the summer.

The survival of the thalli of *F. guiryi* at Punta Calaburras in midsummer correlates with the overall mean NAO value recorded from April to June, with presence clearly favoured under positive NAO, whereas the alga did not survive or did not develop from microscopic stages under negative overall mean NAO values from April to June. It must be taken into account that NAO is an atmospheric "teleconnection" pattern affecting the climatic conditions in the North Atlantic region, and the derived NAO index is a measure of the strength of the sea level air pressure gradient between Iceland and the Azores, which integrates several climatic variables (e.g., water temperature, prevailing wind direction and speed, and precipitation). In the Alboran Sea, a significant relationship has been found between the negative NAO phase and an increase of SST, possibly through increase in run-off (*Báez et al., 2013*). Consequently, it is possible to consider a direct effect of SST on *F. guiryi* by NAO. The sequence of events for the persistence until midsummer of *F. guiryi* thalli at Punta Calaburras could be the growth from microscopic, cryptic stages, as well as the arrival of few-celled embryos originating from the populations located on the shores of the Strait of Gibraltar in winter-spring; then, there can be growth of young thalli if the SST remains relatively low. However, it could be hypothesized that the survival of the thalli is favoured directly both by NAO, and SST resulting from NAO. Thus, positive phases of the NAO during April and June produce dry springs and clear skies. On the other hand, the position of the AJ is variable (*Vargas-Yáñes et al., 2002*; *Macías, García-Gorriz & Stips, 2016*) with a north-south migration pattern (*Sarhan et al., 2000*). The speed of the incoming AJ increases at low pressure over the western Mediterranean (*García-Lafuente et al., 2002*), and decreasing Mediterranean sea-level has been related to positive NAO index (*Timplis & Josey, 2001*). Increasing velocity enhances the Coriolis force and separates the

AJ from the Spanish coast, facilitating the upwelling of cold Mediterranean water (14 °C to 17 °C) at the Spanish coast and consequently allowing the survival of *F. guiryi* thalli, as was suggested by *Lourenço et al. (2016)* in other locations where the alga appears associated with an upwelling. In contrast, under negative NAO index, the AJ velocity might decrease and the Western Alboran gyre (see Fig. 1A), characterized by warmer water, migrates northward and may reach the coast at Punta Calaburras. This increases the probability of short-term periods of very warm water (up to 22 °C) that hinder survival of *F. guiryi* thalli under negative NAO index. It is significant that other organisms do not proliferate on the substrate occupied by *F. guiryi* at Punta Calaburras.

The effect of long-period climate variability such as the Atlantic Multidecadal Oscillation (AMO) on the growth of seaweeds has been recently recognized (*Halfar et al., 2011*), but we cannot correlate our data to this phenomenon because our time series for *F. guiryi* is relatively short.

When the thalli of *F. guiryi* at Punta Calaburras proliferate until midsummer, they do not show evidences of physiological stress under submerged conditions, when compared to the population at Tarifa. However, the photosynthetic performance of thalli at Punta Calaburras is clearly less efficient in air than the counterpart population at the Strait of Gibraltar. The narrower, and overall lower tidal range at Punta Calaburras in comparison to Tarifa (see Fig. 2) ensures that the thalli of the former population remain almost permanently hydrated, whereas the Tarifa thalli experience true submersion-emersion cycles, and consequently are better acclimated to air exposure. It should be noted that the substrate available for the alga at Punta Calaburras is scant, as the rocks do not protrude from the mean sea level more than 30 cm. In contrast, the substrates at Tarifa emerge more than 3 m above mean sea level. Thus, we can hypothesize that the thalli at Tarifa lose water in air more slowly than those at Punta Calaburras. However, the difference in water economy and photosynthetic performance when the thalli are exposed to air is not reflected in the same way in the development instability. The population from Tarifa showed a significantly higher standard error of the estimate of the regression derived from the box-counting method than the population from Punta Calaburras. This result suggests that the former population integrates a higher stress during ontogeny than thalli developing at Punta Calaburras. More studies are necessary to clarify the relationships between stress estimates based on rapid physiological responses and integrative responses during ontogeny.

Summarizing, the connection between the climate variability due to the NAO seems to modulate the midsummer survival of macroscopic thalli of the isolated population of *F. guiryi* in Punta Calaburras, with the presence of the thalli favored in midsummer if the overall mean NAO value from April to June is positive. In this case, the growth of thalli does not reflect physiological or integrative stress in comparison to the neighboring populations, with the exception of water and carbon economy.

## ACKNOWLEDGEMENTS

The suggestions and criticisms of two anonymous reviewers as well as Dra. Ester A. Serrão are gratefully acknowledged. Dr. Eric C. Henry kindly revised the English style and usage.

Herbarium MGC (Universidad de Málaga) kindly provided us the scanned images of the sheets of *Fucus guiryi*.

### Funding

This work was supported by the project "Variabilidad funcional y dinámica de las respuestas al cambio climático de bosques marinos (MARFOR)" from the Ministerio de Economía, Industria y Competitividad, (Acciones de Programación Conjunta Internacional, PCIN-2016-090). The funders had no role in study design, data collection and analysis, decision to publish, or preparation of the manuscript.

### Grant Disclosures

The following grant information was disclosed by the authors:
Variabilidad funcional y dinámica de las respuestas al cambio climático de bosques marinos (MARFOR).
Acciones de Programación Conjunta Internacional: PCIN-2016-090.

### Competing Interests

The authors declare there are no competing interests.

### Author Contributions

- Ignacio J. Melero-Jiménez analyzed the data, wrote the paper, prepared figures and/or tables, reviewed drafts of the paper.
- A. Enrique Salvo analyzed the data, reviewed drafts of the paper.
- José C. Báez and Elena Bañares-España performed the experiments, analyzed the data, reviewed drafts of the paper.
- Andreas Reul analyzed the data, contributed reagents/materials/analysis tools, reviewed drafts of the paper.
- Antonio Flores-Moya conceived and designed the experiments, performed the experiments, analyzed the data, contributed reagents/materials/analysis tools, wrote the paper, prepared figures and/or tables, reviewed drafts of the paper.

### Data Availability

The raw data has been provided as a Supplemental File.

### Supplemental Information

Supplemental information for this article can be found online at http://dx.doi.org/10.7717/peerj.4048#supplemental-information.

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
