# Peer review of "North Atlantic Oscillation drives the annual occurrence of an isolated, peripheral population of the brown seaweed Fucus guiryi in the Western Mediterranean Sea"

_PeerJ, doi:10.7717/peerj.4048_

## Round 0.1 · original submission · Major Revisions

· Academic Editor

Major Revisions

The points raised by the first referee require a major re-thinking of the paper. There seem to be unvalidated assumptions about species' dispersal when in fact the species is always present and just varies in abundance seasonally. Also, is it possible that there are two species within what is know as this species at present? I hope you can re-write the paper to account for alternative interpretaitons and certainly not mislead readers.

Reviewer 1 ·

Basic reporting

I outline specific issues to be fixed below:

-This manuscript uses clear language throughout, although it would benefit from an additional readthrough by a native English speaker.

-The last paragraph of the introduction should be removed -- it is more suitable for inclusion in the abstract, or perhaps the discussion.

-In lines 160-161, the authors state that the 'AO is characterised by a merdional dipole in sea level atmospheric pressure between polar regions and mid-latitude'. This is the NAO, not the AO. The latter half of the sentence '...and could be interpreted as the surface signature of modulations in the strength of the polar vortex aloft' is verbatim from the abstract of Thompson & Wallace 1998 and should be rephrased.

-Throughout this manuscript, the authors write out and explain equations that are standard in the field. (A particular example of this is the approx. 1 page devoted to explaining chlorophyll fluorescence on lines 271-289.) Replacing this text with one-two sentences and a citation for chlorophyll fluorescence would suffice. Similarly, equation 1 and the text following (lines 127-131), equation 2 and the lines following (137-142), etc. could be removed without losing any clarity in the manuscript.

-In the figures (3, 5, 6) please use different shaped symbols in addition to different colors to distinguish among your results. This is necessary for any color-blind readers, as well as printing in black&white.

-In figure 4, please clarify what the grey and orange boxes represent. And, please make the shading/hatching different between the two colors (see color blind readers, above).

Experimental design

The method of sampling for Fucus guiryi presence/absence needs to be described in the Methods: how was it checked? Quadrats and transects? Visual searching over a defined period of time?

Validity of the findings

The most significant issues with this manuscript are the assumptions made about the basic biology of Fucus guiryi, which then impacts the interpretation of the authors' findings in the abstract, conclusions, etc. The authors assume that the presence of F. guiryi in the summertime is solely due to the transport of thalli and embryos from neighboring populations to the west. However, it is always present at their key study site in the winter. (It is not stated how the winter populations appear.) No mention is made of the reproduction of F. guiryi and the possibility of microscopic stages/remnant holdfasts of macroscopic stages remaining even when juvenile or adult thalli are not present (and thus it appears to the naked eye that F. guiryi is gone). The authors could have tested specific issue for this by putting out settlement plates at their sites during different years -- any F. guiryi that appeared during a summer season would have to be due to the settlement of new individuals. Also, the fact that F. guiryi was present _every_ winter of the survey indicates the low likelihood that this population is solely dependent on recruitment from a western source population. In a broader context, it is possible that the appearance of summer F. guiryi is correlated with the NAO...not because the NAO brings F. guiryi to this site, but instead because it creates favorable conditions for existing F. guiryi to grow into macroscopic sizes. The authors should consult Schiel and Foster 2006 doi:10.1146/annurev.ecolsys.37.091305.110251 for a review of the population biology of fucoids.


The authors state that Tarifa is the 'seeding' population. As this was not conclusively demonstrated/experimentally tested, they should state instead that it is a 'likely seeding' population. (see comments in #1 above)

The authors do not examine or discuss the impact of herbivores on blade shape, which can significantly impact their discussion/results of developmental instability. Some discussion of the impacts of herbivory on blade shape is needed.

·

Basic reporting

MAIN: This paper reports a long (26 years, 1990-2015) time series of records of occurrence of a marine alga at an isolated warm range limit. This is a fantastic opportunity as such a data set is difficult to obtain and has allowed the authors to make a more detailed analysis of causes of variability in presence / absence than what is usually possible. The analyses conducted allowed a correlation of the presence of the species with the overall mean value of the NAO (an index reflected in mean weather conditions) from the spring months. Spring is the main reproductive / recruitment season for the species, This is an important topic that deserves publication. I recommend that it should be published quickly.

Main suggestion 1. The paper could be even further improved if the authors could try to use a better proxy for the local conditions of this very localized study area rather than just the NAO.

Main suggestion 2. The introduction could be more clear in the formulation of hypotheses/questions behind the other measurements conducted. The measurements conducted in July 2011 (radiation, temperature, photosynthetic parameters) are a single point in time relative to the time series and it is hard to related the two things and to understand the aim of these point measurements relative to the goal of the paper. I recommend a better explanation of what question are you trying to answer with these measurements. Also, a comparison conducted in the field, under distinct environments can be due to population differences or environmental differences. So, again, please explain better what is the question.

Main suggestion 3. I recommend a better literature review of other such cases of variability of edge populations of Iberian algae.

3a. First, please compare your results with the very relevant paper on variability in Fucus guiryi at its southern limit by Carla Lourenço et al (2016). It reports the dependence of this species on specific local climatic effects namely the upwelling conditions in isolated spots. It also reports the southern limit of Fucus guiryi.
Lourenço CR, Zardi GI, McQuaid CD, Serrao EA, Pearson GA, Jacinto R, Nicastro KR (2016) Upwelling areas as climate change refugia for the distribution and genetic diversity of a marine macroalga. Journal of Biogeography 10.1111/jbi.12744

3b. Likewise, there are other papers for the Iberian Peninsula that have studied long-term variations of species at their range edges, not just for the warm edges in the south but also for the warm edges in northern Spain, where surface seawater warms towards East.

Other suggestions:

4. The index of developmental instability is interesting but can be highly affected by grazing rather than climate stress. Fucus guiryi are extremely consumed by grazers, namely Sarpa salpa fishes that get to it at very high tides. Did this not affect your measurements? Was this population located so high on the shore that fish could never get to it thus not affecting thallus development? This is a tradeoff with the physical stress factors that increase further up on the shore.

5. The chlorophyll fluorescence parameters are widely used in the literature and do not need to be described in detail, with equations, in this paper that simply uses them.

6. Disappearance and recovery in the same site, so isolated from the remaining populations, suggests that the population might have kept microscopic latent stages (embryos of Fucus can remain undeveloped but alive if conditions are limiting) during unfavourable years, that recovered the population afterwards. This hypothesis has not been addressed in the paper. See for example our discussion of such issues for an iberian kelp species (despite the differences between these in being subtidal and having gametophytes): Assis J, Coelho N, Alberto F, Valero M, Raimondi P, Reed D, Serrao EA (2013) High and distinct range edge genetic diversity despite local bottlenecks. PLOS ONE 8(7): e68646

7. Is fig. 4 really necessary? It can be simply reported in the text.

Experimental design

no more comments (see above)

Validity of the findings

no more comments (see above)

Comments for the author

no more comments (see above)

Reviewer 3 ·

Basic reporting

I found the manuscript well structured, the analyses adequate and sound, the conclusions of an environmental control for species occurrence supported by the data. Text in general was also well written and ideas very easy to follow.

Experimental design

No comment

Validity of the findings

I only have one concern (see also a few minor comments below) relating to the authors interpretation of the nature of the marginal Calaburras population. The authors argue throughout the text that F. guiryi is present every year at least during the winter and that some unfavourable years the population does not survive to midsummer. The sole explanation proposed for the recurrent recruitment during the winter is the putative transport of drifting thalli and unattached embryos from neighbouring Gibraltar populations (some 80km away) by the Atlantic Jet (lines 59-61; 396-401; 424-426). Genetic data shows that Fucus (and canopy-forming brown seaweeds in general) are very poor dispersers at km scales (e.g Neiva et al. 2012). There is a much more parsimonious explanation for the recurring appearance of F. guiry at the very same site – the presence of cryptic, hardy microscopic stages that survive unfavourable months before developing again during milder months or after catastrophic events (Carney & Edwards 2006; Creed et al 1996). Local(not immigrated) few-cell zygotes (something like a seed bank) or remnants of thalli (as in many cryptic perennial Cystoseiracea that lose most biomass during the unfavourable winter months) are good candidates, and both avoid the need for a constant supply of propagules from a 80 km distance source. This hypothesis should be at least discussed as an alternative to the immigration scenario.

Comments for the author

Minor comments:
line 48: F. guiryi occurs as far south as Dahkla in the Western Sahara (Lourenço et al. 2016, also discusses the importance of upwelling and lower SST in this African range)

line 49: “sometimes a population develops at Punta Calaburras”. If I understand correctly this is not very clear in the sense that the population is present throughout the years at least during some months and not missing some years

line 52: “(sub F. spiralis and F. spiralis var. platycarpus)”. I don’t understand what these refer to – subspecies? To avoid confusion over the focal species it should be removed or clarified. Same in line 62.

line 67: “university of AFM”- in full if not defined earlier

line 73: “presence at Punta Calaburras – add something like “during/until mid-summer” to avoid confusion. Same in line 394 – add “mid-summer” before “occurrence”

line 166: it could be explicit the institutional source of the data (NOAA?) for AO and NAO, as for SST, instead of just the link

line 423: “persistence until midsummer” would be clearer than “success of occurrence in midsummer”, since the species is there already during the winters. Same in 445 – “proliferates until
midsummer” may be more accurate.

line 449: In calaburras the tidal range is narrower but also overall lower. Stating something like “the narrower, [and overall lower] tidal range” reinforces the differences discussed immediately after

Figure 1: correct “Westarn” Alboran in legend

Figure 2 and 6: clarify UTC in legend

Figure 3: correct “ornage” in legend


References
Neiva et al. 2012 Drifting fronds and drifting alleles: range dynamics, local dispersal and habitat isolation shape the population structure of the estuarine seaweed Fucus ceranoides.

Carney & Edwards 2006 Cryptic processes in the sea: a review of delayed development in the microscopic stages of marine macroalgae

Creed et al. 1996 The development of size structure in a young Fucus serratus population

Lourenço et al. 2016 Upwelling areas as climate change refugia for the distribution and genetic diversity of a marine macroalga

---

## Round 0.2 · Minor Revisions

· Academic Editor

Minor Revisions

While the referees make positive recommendations it is dissapointing that they find so many revisions still necessary, to correct mistakes and more correctly phrase and articulate the paper. Please be very careful to correct all of these and check for any others they may have overlooked if you decide to revise the paper further.

Reviewer 1 ·

Basic reporting

This revised manuscript is much clearer to read than the original version -- thank you! However, there are still grammatical English errors throughout the manuscript that need to be fixed. I highlight some, but not all, in this review. (e.g. the third line of the Abstract reads 'The presence of the alga at Punta Calaburras is supposed could be due to the growth of....' )

The authors should remove the last part of the last sentence in the Abstract 'an aspect practically unknown before'.

Lines 56-57 should be rephrased - unclear.
Line 67: remove 'uninterrupted'
Line 70: remove 'must' -- this has not been experimentally confirmed.
Remove the two sentences that start on lines 136 and 138 - they are not needed in this manuscript.
Line 143: fix the ...
Lines 172-174: remove these first two sentences, as they do not belong in the Methods section.
Line 184: change 'image' to 'images'
Line 232: change 'cultured' to 'placed'
Line 363: replace 'last' with 'recent'

Experimental design

The authors have not yet clarified how the sites were surveyed for the presence/absence of the algae. Was the entire site visually surveyed? Were quadrats/transects used? The presence of small macroscopic stages may require careful surveying techniques, and this is not clear from the text written.

The authors need to describe how the thalli were selected for the developmental instability analysis. Was this random? Haphazard? Both Random and Haphazard selections of individuals have specific ecological criteria. The authors need to demonstrate how bias was avoided when thalli were selected. Similar comment for line 230 in the fluoresence Methods section.

The authors should change dry weight to dry mass, as they were recording mass, not weight. Also, the equation in line 243 is wrong. The denominator should be fresh mass, not dry mass. Please note that this equation is no longer Equation 5. Similarly, Equation 7 is not equation 7 anymore.

Validity of the findings

Line 278-279: rephrase this sentence. A trend of what through time?

Binary logistic regressions (line 280-294) are not linear. I am not sure where the equation came from in line 285, but it does not represent the model output that is shown in Figure 3. This section needs to be rewritten to reflect this.

The authors state that there was not a difference in developmental instability between the two sites, but they had a p=0.04 value for difference by location. This is significant (as the authors do not state that they used a different p value as a cutoff for significance), but the authors treat it as if it were not. They also do not state which site had the higher developmental instability values.

Figure 2 should also show the tidal range for Pt. Calaburras.

Comments for the author

The first paragraph of the Discussion is repetitive from the Introduction. It should be shortened substantially to focus on how the new results from this work fit with current ecological theory on peripheral populations. Additionally, the second part of this paragraph (~371-401) should be significantly shortened for clarity. The authors should simply discuss that the population most likely re-emerges from existing microscopic stages already present at Punta Calaburras, and that dispersal from the Strait of Gibraltar is possible, albeit unlikely.

Remove 434-437 - these sentences are not relevant for this manuscript.

Change the Discussion text regarding the developmental instability results.

·

Basic reporting

The authors have addressed the reviewers concerns adequately.

Experimental design

The authors have addressed the reviewers concerns adequately.

Validity of the findings

The authors have addressed the reviewers concerns adequately.

Reviewer 3 ·

Basic reporting

This is the 1st revision of the manuscript by Melero-Jiménez et al. investigating the climatic and oceanographic correlates of mid-summer persistence of a range-edge population of the seaweed F. guiryi and its comparison (developmental and physiological traits) with a less marginal population.

Globally I found the revision adequate as most comments raised were addressed. I only have a few additional minor comments/suggestions that the authors can address without the need for an extra round of revision.

Some ambiguity remains in some sentences concerning the seasonal nature of F. guiryi at Calaburras – it should be clear to the reader that macroscopic F. guiryi is always detected in the winter and either these individuals persist until mid-summer or they disappear, depending on the years. Specifically, that midsummer occurrence is related to survival of winter alga and not recruitment (this happens all years during winter). The authors can consider some suggestions (see minor comments below) that hopefully improve clarity.

Experimental design

R1 mentioned in the first review that it was not clear how presence/absence had been assessed (transects, quadrats, searching haphazardly, other?). This information may be relevant for the readers and should be made available.

Validity of the findings

My personal view (based on available literature) is that migration from Gibraltar is much less plausible than local recruitment. In any case, long-distance (non-local) dispersal by kelp and fucoid seaweeds is much more often mediated by drifting, reproductive thalli (that release their fertilized zygotes in situ after being washed ashore) than by direct travel of spores or zygotes, that in the case of Fucus are actually negatively buoyant (i.e. they sink). This alternative vector of long-distance dispersal (drifting reproductively thalli) from Gibraltar may be worth mentioning.

Comments for the author

Minor comments/suggestions
line 24 – rephrase and correct “is supposed could be due to the growth of resilent, microscopic stages as well as influence of the permanent Atlantic jet”

line 43 – the first sequence is too long; it can be break in two sentences or at least use a coma. It may also be more correct to add “being more abundant (towards warmer range edges) in areas characterized by upwelling”

line 47 – the “known” southern limit would be more correct

line 51, 61- the presence of F. guiryi (named as F. spiralis and F. spiralis var. platycarpus; see Zardi et al., 2011). F. guiryi (named as F. spiralis; see Zardi et al., 2011 was found in 1987…
It may be clearer as “previously named” or “previously/originally identified as” F. spiralis etc…

lines 56 – upwelling causes lower SST and allows persistence in otherwise too hot areas, but this effect does not necessarily have to do with modern climatic change. It may be more correct to rephrase as “in agreement with the hypothesis of upwelling providing thermal refugia/relief for F. guiryi”.

line 67 – “field teaching at the university of AFM”. For clarity it could be added “at the university of the corresponding/senior author AFM”, otherwise readers (like me) may think it is the university abbreviation.

line 70 – Seasonality appears to be exclusive of Punta Calaburras. This could be reinforced as “in contrast to the nearby perennial populations in the Strait of Gibraltar in the Strait of Gibraltar AND the remaining range.

line 87 - replace “population” by “populations”

line 189 – correct “can influences”

line 372-374: this phrase is a bit confusing, please rephrase

line 375: replace “occurrence of thalli” by “re-appearance of thalli”

line 387: “Another, alternative hypothesis is that recurrent re-appearance of F. guiryi at Calaburra is achieved by…” “unfavourable summer months”…

392: local microscopic stages (recruits, holdfast remnants) at PC ca work as a seed bank

393: With our data, we cannot discern whether the recurrent re-appearance of the alga during the winter is due to migration or local recruitment, or both. Taking into…

401: correct F. guiyi

404: “involved in the annual occurrence of the alga” by “involved in the persistence of macroscopic thalli during the summer”

line 421: replace “occurrence” by “survival”

line 461: replace “modulate the occurrence of the annual survival of” by “modulate the mid-summer survival of macroscopic thalli of”

line 470: Dra Ester Serrão

---

## Round 0.3 · Minor Revisions

· Academic Editor

Minor Revisions

I gave the MS a read and found some editorial mistakes (misspellings etc.). Please consider these and give it another proof read because the PeerJ editorial office does not do detailed editorial corrections like older journals might. The web page would only let me upload a PDF but the Editorial Office will forward you the PDF.

---

## Round 0.4 · Minor Revisions

· Academic Editor

Minor Revisions

The abstract has three spelling mistakes all highlighted by spell checker in MS Word: peristence; occurences; fractral.

This is just the first page. Please can you give it a more careful proof read before finalising?

---

## Round 0.5 · accepted · Accept

· Academic Editor

Accept

Thank you for re-checking the text and a careful proof read.